# How Loan Bank of Assistive Technology Impacts on Life of Persons with Amyotrophic Lateral Sclerosis and Neuromuscular Diseases: A Collaborative Initiative

**DOI:** 10.3390/ijerph18020763

**Published:** 2021-01-18

**Authors:** Thais Pousada, Jessica Garabal-Barbeira, Cristina Martínez, Betania Groba, Laura Nieto-Riveiro, Javier Pereira

**Affiliations:** 1TALIONIS Research Group, Research Center of Information and Comunication Technologies (CITIC), Faculty of Health Science, Universidade da Coruña (University of A Coruña), 15006 La Coruna, Spain; b.groba@udc.es (B.G.); laura.nieto@udc.es (L.N.-R.); Javierp@udc.es (J.P.); 2Galician Association of Amyotrophic Lateral Sclerosis (AGAELA), 15670 Culleredo, Spain; jessicagarabal@gmail.com (J.G.-B.); info@agaela.es (C.M.)

**Keywords:** assistive technology, amyotrophic lateral sclerosis, Neuromuscular Disorders, occupational therapy, outcome measures

## Abstract

(1) Background: The study is focused on the implementation of outcome measurement tools to assess the impact of an assistive device from a loan bank in the lives of people with Amyotrophic Lateral Sclerosis and Neuromuscular Diseases. The secondary purpose is to analyse the correct matching between the person and technology, derived from the counselling of an occupational therapist. (2) Methods: Cross-sectional study. The sample was formed by 28 people with rare neurodegenerative disorders. A specific questionnaire, the Psychosocial Impact of Assistive Device Scale (PIADS), and the Matching Person and Technology (MPT) tool were applied to collect the data. (3) Results: The dimension of the PIADS with the best score was competence, and the variations according to gender were not remarkable. The three dimensions of the PIADS (competence, adaptability, and self-esteem) were correlated positively between them and with the mean score of the MPT tool (*p* < 0.01). The type of assistive technology (AT), diagnosis, and correct match between person–technology are the main factors that condition a positive impact. (4) Conclusions: The results noted the importance of assessing the needs, demands, and contexts of people with rare neurodegenerative diseases to prescribe the best AT. Loan banks of AT have to be considered a valid service that complements their lack in public health services.

## 1. Introduction

The World Health Organization (WHO) estimates that 1 billion people globally need assistive technology (AT), such as wheelchairs, prostheses, glasses, and hearing aids. However, only 1 in 10 people have access to the assistive products they need [1]. Assistive products, such as wheelchairs, glasses, and hearing aids, have the potential to enable persons with disabilities, older people, and persons with non-communicable diseases to overcome functional limitations; to lead a healthy, independent, and dignified life; to access employment and education; and to participate in society [2,3].

Rare neurodegenerative diseases, such as amyotrophic lateral sclerosis (ALS) and neuromuscular diseases (NMDs), have an important impact on the independence of the affected person. Their progressive advance decreases the functional capacity of these people, and they can require more support over time, whether it be from assistive devices and/or the caregiver. So, this help is crucial to maintain a minimum of personal autonomy [4,5,6]. Moreover, the caregiver has to assume an important role as the vital support to the affected person, so the burden can appear, and in fact, exist on the relative that offers this care [7,8,9].

An important part of the efforts toward universal coverage for assistive devices involves building a national research capacity to generate a strong evidence base both for the real demand for wheelchairs in the communities and to prepare for the necessary investments in infrastructure and human resources [1].

One of the most important barriers preventing the inclusion of AT within public health policies, as well as undermining access, is the lack of evidence on the benefits and potential of AT. Understanding the impact of AT on people’s well-being, quality of life (QoL), inclusion, participation, empowerment, and social and economic status is key to inform policy development, guide public investments, and mobilize resources [1,10,11,12].

Taking this into account and to guarantee normal access to the main ATs, some non-governmental organization (NGO) such as the Galician Association of Amyotrophic Lateral Sclerosis (AGAELA) have created a “loan bank of assistive technology” [13]. This service works like a library and offers AT to the members of the AGAELA and those with NMDs who are part of the Galician Association against Neuromuscular Diseases (ASEM Galicia), which has an agreement with the first NGO. In other words, a person with ALS or NMDs that needs any device for mobility, communication, or rest applies for it by asking the occupational therapist (OT) at the AGAELA. The professional assesses the needs of the person and approves the loan. This service is free for all users of the NGO and their families. On the other hand, if any individual has any assistive device that he/she no longer uses or no longer needs, he/she can donate it to the loan bank. So, the number of devices to be loaned has increased over time.

At present, the loan bank has 303 devices (that can be viewed on its website: https://agaelagalicia.wixsite.com/bancoproductosapoyo/catalogo). From its beginning in 2014, a total of 149 devices have been loaned to 74 beneficiaries. This is because one person can apply for more than one AT. Some users have three or four devices or can change them over time, based on their current needs.

Several studies have shown that the most frequent and influential factors during the process of prescription and subsequent use of AT is the lack of matching between the person and prescribed assistive device [14,15]. This problem may lead to the abandonment or non-use of the AT. The non-use of devices has clear causes and obvious consequences not only on the QoL and the autonomy of the person with disability, but also cost–benefit effects [16]. To achieve an increase in the successful use of AT and, therefore, to reduce the probability of being abandoned or not fulfilling its initial function, it is essential to carry out an adequate assessment of the user’s needs with reliable measurement tools to improve decision-making for the prescription and adaptation of AT.

The outcome measure is a set of considerations and tools to determine if a service, product, or device allows the achievement of the goals for which it has been created, under criteria of efficacy and effectiveness [17]. A revision done by the ATOMs Project identified 22 models of outcome measures related to AT and 14 tools or instruments to assess the results of that device, such as the psychosocial impact of the device on the QoL, satisfaction with the AT, or level of matching between the person and technology [18]. In Spain, the research related to outcome measures in AT is deficient, because there are not many specific measurement instruments that are validated in the Spanish population. The application of these tools has been reflected only in two research projects [16,19,20].

Due to the experience of AGAELA managing the loan bank of AT and its specific support to people with ALS and NMDs, authors have raised this research. Therefore, the present study is focused on the implementation of outcome measurement tools to assess the impact of an assistive device on the lives of people with rare neurodegenerative diseases in order to improve the efficacy, effectiveness, and real utility of the loan bank of AT.

The main objectives of this study were (1) to determine the impact and benefits of the assistive device received from the loan bank and (2) to analyse the correct matching between the person and technology, which is derived from the counselling of an OT.

## 2. Materials and Methods

### 2.1. Design

This is a cross-sectional study that has been applied after the prescription and provision of AT.

The period to obtain and register the data was from March to July of 2020. However, the process of prescription, delivery, and use of AT from the user includes the time from the opening of the loan bank (in April 2014) to December 2019. The service, as commented before, is provided by a NGO formed by people with ALS (the AGAELA) and directed not only to its members, but also to the members of a second NGO, the ASEM Galicia, formed by people with NMDs. In other words, the ATs are loaned to the members of both NGOs, thanks to the agreement between them.

Before implementing the research, the presidents of both NGOs were consulted, and they gave the authorization to perform the study, after getting approval from the Galician Ethical Committee.

### 2.2. Participants

The sample was formed by people with rare neurodegenerative disorders who are members of two NGOs in Spain, one formed by people with amyotrophic lateral sclerosis, and the other formed by people with neuromuscular diseases. The first one (the AGAELA) is the NGO that manages the loan bank of AT for both associations. So, the recruitment of participants has been conditioned by this context.

To select the participants, several inclusion criteria were defined: (1) A diagnosis of ALS or NMDs, (2) a moderate or medium level of dependence, (3) receipt of any AT from the loan bank between 2015 and 2019, and (4) signed informed consent to participate in the study. All individuals that met the inclusion criteria (74 beneficiaries of the loan bank) were invited to participate in the study. A total of 30 people participated in the survey. However, two of them weren’t included in the pool of total participants, because they did not complete the whole survey.

Finally, the sample was formed by 28 persons: 23 with ALS and five with NMDs.

The study was done in the region of Galicia, in Spain, where the two NGOs of people with rare neurodegenerative diseases (the AGAELA and ASEM Galicia) are working and offering the services of a loan bank of AT.

### 2.3. Variables and Instrument Measures

Specific questionnaire: To collect the main sociodemographic data, a simple questionnaire was created to be applied among the participants. The variables recorded through this questionnaire were: Age, gender, city of residence, context of residence (urban, semi-rural or rural), type of AT, reason for applying for the loan of an assistive device, and who has applied for the AT (person with ALS or NMD, caregiver, or professional). The participant also marked if the device was returned or whether he/she was still using it.Psychosocial Impact of Assistive Technology Scale (PIADS): The PIADS was developed to address the need for QoL-related outcome indicators relevant to the use of AT. It has been used to assess the impact of a variety of ATs in various populations [20,21,22,23,24]. It is a self-report questionnaire that assesses the effects of having access to an AT on functional independence, well-being, and QoL [21]. It consists of 26 items grouped into the following three subscales: Competence (12), adaptability (6), and self-esteem (8). The scale captures both positive and negative effects using a scale with ranges from −3 (maximal negative impact) to +3 (maximal positive impact); no effect is scored 0. The validity, reliability, and sensitivity of the PIADS have been described in several publications as being good to excellent [21,25], and a Spanish version is available [26]. Additionally, several studies have shown that the PIADS can be answered by the caregiver [27].Matching Person and Technology (MPT) tool: The MPT tool is a model based on the factors of person, milieu, and technology, and it considers important aspects of these domains in the assessment through a process divided into several steps in a user-focused assessment of AT. The domains of the MPT tool are important factors to consider for anybody waiting to receive any assistive device. So, the implementation of models, such as the MPT tool, will contribute to improving the quality of AT services. “The MPT is a set of questionnaires developed to identify the AT that best suits the user’s goals and preferences, technology features, and environmental support” [28]. In the case of this study, authors selected one of those questionnaires. So, the Assistive Technology Device Predisposition Assessment (ATD PA) device form was applied. This assessment may be completed for one AT being considered or for several competing devices, and it has been designed to assess the level of matching between the person and AT used in different contexts and activities. This part of the questionnaire (device form) asks for the consumer’s views of competing choices, a rating on the consumer’s satisfaction, and various other important issues, which have proved to be vital information. The ATD PA was useful as a follow-up assessment in deciding where to help the consumer be more successful. It has 12 items, and each of them can be scored through a Likert scale from 1 (never) to 5 (always). The total score is calculated with the sum and the mean of the score for individual items [29]. The MPT instruments have been shown to have very good reliability and validity, and it has been validated for use in Spain [30,31].

### 2.4. Procedure

All participants that had received any AT from the loan bank and met the inclusion criteria were invited to participate in the research.

Once the information was sent to the candidates involved in the research and they gave their consent, the participants received a link to access the questionnaires/survey online. In case the participant didn’t have access to the Internet or didn’t have the opportunity to answer the questionnaires online, a researcher from the team (the professional) called her/him to apply the instruments through a phone interview.

In order to facilitate the participation of people with these diseases, their caregiver could help them during the access and process of filling out the questionnaires through the online link, marking the answer given by the person using the AT.

Although some people could have applied for two or more devices, participants were asked to designate the one that they considered most important or that would have the most use.

### 2.5. Data Analysis

All data were gathered by RedCAp, a secure web application for building and managing online surveys and databases. It is specifically geared to support online or offline data capture for research studies and operations. Those data were analysed using the SPSS v.24 program. The statistical assumption of the normal distribution of the data was confirmed after applying the Shapiro–Wilk test. So, parametric tests were used during the analysis of data.

Descriptive analysis was done with expression of the data as means and standard deviations (SDs) or frequencies and percentages, as appropriate. With respect to inferential analysis, the significant statistical relationships were highlighted. In order to establish a correlation between quantitative variables, Pearson’s linear correlation coefficient was calculated. Taking into account the normal distribution of the sample and after checking that the homogeneity of variance was met, Student’s paired samples t-test was applied comparing average values between the two groups, while a one-way analysis of variance (ANOVA) parametric test was applied comparing average values between more than two groups.

Finally, hierarchical linear regression analyses for each dimension of the PIADS were performed after controlling for relevant covariates. Covariates with significant influence (obtained from the inferential analysis) were sequentially entered into the model to examine the relative weight of each variable. That is, hierarchical linear regression was used to determine if independent significant variables explained a statistically significant amount of variance in the results of each dimension of the PIADS, after accounting for all other variables.

The level of statistical significance was set to *p* < 0.05.

### 2.6. Ethical Concerns

The main principles of ethical guidelines for human research were followed. The data of participants were guaranteed anonymity and confidentiality, and participants were informed that their participation was voluntary. Informed consent was obtained from all individual participants included in the study. They could withdraw from the study without any consequences for future services at any time. The study was approved by the Galician Ethical Committee, with the reference number 2019/215.

The funders had no role in the design of the study; in the collection, analyses, or interpretation of data; in the writing of the manuscript; or in the decision to publish the results. All procedures performed in studies involving human participants were under the ethical standards of the Research Ethics Committee of Galicia and the 1964 Declaration of Helsinki and its later amendments or comparable ethical standards. This article does not contain any studies with animals performed by any of the authors.

## 3. Results

A total of 28 persons were included in the final sample, with the median of age 59 (ST = 13.00). Table 1 shows the main characteristics of the sample. The main type of AT requested was for rest (*n* = 10; adjustable beds), followed by those for mobility (*n* = 7; electric wheelchairs (3), walkers (3), and a wheelchair with positioning system (1)). The other categories of loaded assistive devices were AT for transfers (*n* = 5; hoists), bathing aids (*n* = 5, swivelling bath seats (2), swift shower chairs (2), and a shower commode chair (1)), and AT for communication (*n* = 1, a Tobii eye).

The main reason to apply for any device in the loan bank was the progression of the disease, which usually reduces the level of general mobility. Additionally, all requested devices were not included in the national catalogue of AT available through the Public Health Service.

In order to determine the impact of AT on the lives of people with ALS or NMD, the PIADS was applied. Table 2 shows the main results of the obtained score and grouping by the person who answered the questionnaire. The dimension with the best score was competence (*m* = 0.6), with the self-esteem dimension being the worst (*m* = 0.49). The table shows that the score obtained when the survey was answered by the caregiver was lower than the results given by the person affected by ALS or NMD.

The ATD PA device form (from the Matching Person and Technology model) was applied to determine if there was a correct match between the affected person and the loaned device. In this case, the mean score for the 12 items was calculated for all participants. Table 2 shows that the mean was 3.94 (ST = 0.73) out of 5, so it can conclude that the match “person-AT” is quite good. In this case, the score obtained taking into account the individual that covered the survey was not remarkable, but it was a little higher when the professional answered it. The item of the form with the highest score was “The support, assistance and accommodations exist for successful use of this device” (*m* = 4.5), while the item with the lowest mean was “The device fits in all desired environments (car, living room, bathing area, etc.)” (*m* = 3.42).

The inferential analysis was done to determine the possible influence of the variables obtained in the specific questionnaire on the scores of the PIADS and ADT PA. Table 3 summarizes only the variables with a relevant influence on the scores of competence, adaptability, and self-esteem.

The Pearson test applied with the quantitative variables determined that there were significant correlations between age and the dimensions of competence (Rho = −0.45; *p* < 0.05) and adaptability (Rho = −0.44; *p* < 0.05). In other words, the older the age is, the less positive impact (in terms of competence and adaptability) the assistive device has on the person’s life.

Additionally, the three dimensions of the PIADS are correlated positively between them and with the mean score of the ATD PA (*p* < 0.01). In this sense, to check the reliability of the PIADS, the α Cronbach test was applied. In this case, the scores of the three dimensions were the elements of the analysis, and the result for α was 0.91. In other words, we have very good reliability of the PIADS, with the α value very close to 1.

The qualitative variables were also combined with the PIADS and ADT PA scores, applying the t-test or ANOVA, as appropriate. After the analysis, it was concluded that variables such as the gender, the person that answered the survey, the context (urban, semi-urban, or rural), or the situation of having the AT during the study (returned or still using) had no influence on the scores.

Concerning the category of AT (rest, transfers, mobility, bathing, or communication), the dimensions of competence (F = 365.2; *p* < 0.05) and self-esteem (F = 284.3; *p* < 0.05) were significantly influenced by them. The persons that received an adjustable bed (rest) or hoist (transfers) obtained a lower score in those dimensions compared to people using an AT for bathing or communication.

The fact that the AT was requested by the affected person or by the caregiver was a factor that had an influence on the scores obtained in two dimensions of the PIADS: Adaptability (F = 137.4; *p* < 0.05) and self-esteem (F = 258.3; *p* < 0.05), but not for the mean of the ATD PA. In other words, the impact of loaned AT on adaptability and self-esteem was higher when the product was requested by the affected person in comparison with the fact that the caregiver applied for the device. By cons, the match between the person and AT seems to be the same under different types of requests.

The diagnosis was shown to have a relative influence on the scores of the dimensions of competence (*p* < 0.05) and adaptability (*p* < 0.05). In this case, the persons affected by a NMD offered higher scores than those with ALS. The diagnosis had no influence on the match between person and AT (obtained by the ATD PA questionnaire).

Table 4 presents the results of the hierarchical regression model of competence, adaptability, and self-esteem, introducing the independent variables with significant influence on those dimensions, according to Table 3.

In the model of competence, each addition of matching (ATD PA score), type of AT and diagnosis resulted in a significant increase in the changes of variance explained. Age was the variable excluded from the model. Type of AT was the best significant predictor for greater competence (β = 0.61, *p* < 0.01), and model 3 accounted for 71.6% of the variance for competence.

Concerning adaptability, the addition of matching and diagnosis resulted in a significant increase in the changes of variance. The age and the variable of who requested the device were excluded from the model. The matching of the person–technology (ATD PA score) was the best significant predictor for greater adaptability (β = 0.59, *p* < 0.01), and model 2 accounted for 51.1% of the variance for adaptability.

In the case of self-esteem, only one model was created, with the variable score of matching (ATD PA score). The type of AT and the variable of who requested the device were excluded from the model. This model accounted for 57% of the variance for self-esteem.

## 4. Discussion

The present study has tried to capture the impact of the service from a NGO through the loan bank of AT and the impact of the device on end-users. The studied variables have allowed us to know the characteristics that have any influence on the impact of the AT on the lives of people with ALS or NMDs, and if the match between person and technology is correct.

In general, although positive, the impact of the loaned device on competence (*m* = 0.6), adaptability (*m* = 0.56), and self-esteem (*m* = 0.49) was relatively low. It can be explained because the category of the requested AT was more related to facilitating the rest or transfer than with doing activities independently or without the support of the caregiver. Other research, applying the PIADS with a population of ALS or NMDs users of any AT, has also reported a positive impact of the device on the lives of participants [16,32,33,34]. In fact, the type of device showed a significant influence on competence and self-esteem (*p* < 0.05). Like the present study, higher scores were obtained for devices that facilitated communication (eye tracker) [32,35], mobility (powered wheelchair) [16,33], or rest [34].

The fact that starting to use any AT implies that the disease has progressed, very quickly in the case of ALS, with the corresponding loss of functional ability that the person has to assume. So, the role of AT in rapidly progressive rare diseases is more palliative than a promoter of independence r. These results agree with the research of Louise-Bender, Kim, and Weiner (2002), who concluded that “assistive devices can, early on in disease progression, serve as a means of coping with symptoms and limitations, but the fact that device use increases with disease progression indicates that the meanings assigned to AT also subtly change”. For the person with ALS or NMD, this means that “AT use can symbolize further debilitation and/or ultimately death” [36].

On the other hand, the obtained score for the ATD PA device form shows good matching between the person and the loaned device. This finding answers to the second goal, being the OT that manages the loan bank of ATs and that prescribes the most adequate device available from the loan bank, according to the needs of people with ALS or NMDs. Despite having a complete list of assistive devices to loan to any user that needs it, the service is not efficient and effective if it doesn’t have the support of a trained and expert professional. In this case, the OT has the competences to assess the functional skills, needs, and environment, as well as to analyse the significant activities of the person with ALS or NMDs, and based on that information, recommends and prescribes the best assistive device that meets those characteristics [4,6,37]. However, the results obtained in the present study showed that the correct match between person and technology can even be perceived higher by the professional compared with the score of the user. Therefore, the counselling given by the OT must involve the perspective and opinion of the user from beginning. Nevertheless, one of the aspects covered by the ATD PA device form is useful as a follow-up assessment and offers useful feedback for goal-setting and decision-making [29].

After the inferential analysis, it was confirmed that the diagnosis, type of AT, and correct match between person and technology can be predictors of a positive impact of the device on the lives of people with rare progressive diseases, as shown in the model of linear regression. From that, the most important and influential variable is the match between person and device (ATD PA device form score) on the impact of AT. So, again it determines that the success of the service depends directly on a good assessment of the person and his/her environment, a correct prescription of the best AT and complete training in its use done by an expert professional. The manager of the loan bank of AT is the OT, so it is important to consider this discipline in those services that provide and offer the prescription of AT, both in public and private centres, in order to achieve a good quality of services [4,5,6,35,37]. By obtaining information directly from patients, an OT will acquire first-hand knowledge about personal autonomy and the needs of users. These suggestions may help both patients and professionals make informed decisions during the AT device selection process, as well as during the counselling and follow-up stages [6].

By cons, the gender, the context (urban vs. rural), or the perspective of caregiver vs. affected person seems to not have any influence on the impact of the assistive device or the correct match of person–technology.

In the present study, the authors have applied the outcome measures to support the importance of the use of AT on the lives of people with ALS or NMDs through a loan bank of AT. To improve rehabilitation services related to AT and to benefit the final users of these devices, several models of outcome measures have emerged [15,38]. Those models indicated the main factors to take into account in the process of getting a match between the person and AT [18,20] are as follows:The functional problems that AT intends to solve.The characteristics of final users and their needs and priorities.The characteristics of the device that are responsible for its intervention.The context in which AT is applied or used.The expected changes in the state of a user and its context are the results, both short and long term.The impact of AT devices on the individual’s participation in the environment.

In other words, outcome measures are the evaluation process in the provisioning service that is designed to quantify and establish a baseline on something that works (its effectiveness), the group on which it works, and what level of economic efficiency it provides [17,39]. So, professionals prescribing AT have to take into account the need for applying outcome measures to improve their intervention, with an evidence-based process. The research group used the PIADS and the ATD PA device form, because they are the only tools validated in Spain, in addition to the experience in their application and interpretation.

The types of devices included in the present study, and those others that conform to the loan bank of AT of the AGAELA, are not supported or funded by the Public Health Service in Spain. In other words, if anyone needs an automatized bed (very important when the lack of mobility complicates the postural changes at night and very frequently in people with ALS or NMDs), the family has to purchase it. This supposes a strong economical effort on the part the user and his/her family. Resources and services offered by the NGOs of people with disabilities, such as the AGAELA, are the only solution to cover the lack of those from the Public Health Service in Spain.

The WHO General Assembly in 2018 published a resolution on the importance of ATs and services globally (including assessment) and supported the position papers. The main considerations that take into account governments in the provision of AT to all citizens are related to the People, Products, Provision, Personnel, and Policy [1].

Access to appropriate AT is considered a fundamental human right. Addressing the current gap, and providing 900 million people with appropriate technology, is essential to realize the Convention on the Rights of Persons with Disabilities and to achieve the 2030 Agenda for Sustainable Development and its targets related to health, well-being, universal health coverage, economic growth, inclusive education, and sustainable societies. The Resolution WHA71.8 “Improving access to assistive technology” calls on the member states to include AT within universal health coverage and to promote AT policies, programmes, services, and capacity building [3,40]. Moreover, the WHO 13th Global Programme of Work focuses on making a measurable difference in people’s health at a country level and ensuring equity and rights-based approaches to health that enhance participation, build resilience, and empower communities [41].

So, the service of the loan bank of AT is a good alternative and complements providing personal support to users and families, as well as contributing to the reuse of those devices. Taking it into account, it is evident that different stakeholders (between them, governments and user organizations) have to implicate and involve creating an efficient, sustainable, and effective service of loan and reuse programmes of AT.

Other loan banks of AT or similar resources with the same dynamic as the one presented here exist [42,43,44,45,46], but none of them is specifically meant to meet the requirements of people with rare neurodegenerative diseases. In these cases, as their needs change over time, the utility and efficacy of any device can decrease, and new AT has to be implemented in the face of the appearance of new difficulties. So, the loan bank of AT presented here offers the best device for the stage in which the person is and, at the same time, collects those products that will no longer be needed, recycle and repair them, and finally, reuse and reassign them to other users. The benefits of loan banks of AT are clear, contributing to the circular economy, but to get the best service, specific rules and requirements have to be implemented by the organizations, especially those related to policies of management and guarantees of the recipient’s responsibilities [43].

Understanding the many benefits and potential of AT, as well as its impact on people’s lives, is important to improve their quality [2]. However, currently, there are no universally valid or reliable tools or measures that could support member states to capture the impact of AT within their populations. The WHO, together with its partners, is seeking to develop a universal tool to measure the impact of AT on the population’s well-being, QoL, socioeconomic status, human rights, and other similar dimensions. Collecting evidence of measurable changes in these dimensions and the positive impact of AT will greatly contribute to delivering the WHO’s resolution on AT and the 13th Global Programme of Work [41]. So, the present study, which has applied different tools of outcome measure, can constitute a small step to achieve that global goal and offers a good report with results that will help to configure the services of reuse or loan banks of AT.

### Limitations of the Study

The small size of the sample is the main limitation of this study, since, despite 74 people having requested some form of AT from the loan bank since its inception (the total population that met the inclusion criteria), only 28 completed the whole survey. That has implicated that the confidence level and accuracy decreased. Although, it should be noted that there are not any other loan banks of AT specifically for people with rare progressive diseases in our region.

Additionally, it could have some factors or concerns not controlled or not registered that can act as confusing variables. So, in future research, it needs to consider other characteristics, such as time from diagnosis, level of education, or work activity, to get a more complete perspective of the impact of AT (and its consequences) on the lives of people with rare progressive diseases. The fact that the same dataset and performing multiple analyses could decrease the power of significance.

Finally, it is necessary to note that this study has been done in a region of Spain, with its particular public and private health services. So, the results are linked to them, and therefore, these findings have to be interpreted taking into account those contextual characteristics.

So, these features could have an impact on external validity, that is to say, on the extent to which the results of a study can be generalized to and across other situations. Meanwhile, the internal validity and the feasibility were determined by the instrument measures: PIADS and the ATD PA (Device Form), both adapted and validated into the Spanish context. In this research, both scales have shown quite good feasibility after applying α Cronbach test.

## 5. Conclusions

The main conclusions derived from this study are as follows:-The impact of the device loaned from the bank of AT of a regional NGO on the lives of people with rare neurodegenerative diseases (ALS and NMDs) is positive, although moderate.-The main factor that influences that impact is the correct match between person and technology. The matching between the person with ALS or NMDs and her/his loaned device was moderately high. So, the correct prescription done by an experienced professional is vital to achieving this match.-The type of loaned device and diagnosis can act as a possible moderator for the positive impact of AT on the lives of people with rare neurodegenerative diseases. These aspects have to be taken into account in the process of prescription to get the maximum benefit and utility for the affected person and his/her family.-The results of the research noted the importance of assessing the needs, demands, and contexts of people with rare neurodegenerative diseases to prescribe the best AT and to get a high and positive impact of the device on the life of the user.-The OT is one of the main professionals with adequate competences to assess and to offer support in the process of prescribing and providing the assistive devices that people with ALS or NMDs may need.-Loan banks of AT have to be considered a valid service that complements their lack in public health services. The specialization of these banks according to the characteristics of the end-users is essential to guarantee the correct prescription of the device and to get high rates of efficiency.

## Figures and Tables

**Table 1 ijerph-18-00763-t001:** Characteristics of participants.

Variable	*n*	%	
Gender			
Male	15	53.60%	
Female	13	46.40%	
Type of rare disease			
Amyotrophic lateral sclerosis (ALS)	23	82.10%	
Neuromuscular disease (NMD)	5	17.90%	
Living context			
Urban	18	64.30%	
Semi-urban	6	21.40%	
Rural	4	14.30%	
Type of AT			
Rest	10	35.70%	
Transfer	5	17.90%	
Mobility	7	25.00%	
Bathing	5	17.90%	
Communication	1	3.60%	
The device was applied by			
Affected person	13	46.4	
Caregiver	14	50	
Professional	1	3.6	
The device was returned			
Yes	7	25.00%	
No	21	75.00%	
The survey was completed by			
The person without help	10	35.70%	
The person with the help of a caregiver	3	10.70%	
The caregiver	13	46.40%	
The professional	2	7.10%	
	**Mean (ST)**	**Median**	**Range**
Age	58.89 (13.23)	61	51

**Table 2 ijerph-18-00763-t002:** Score obtained for dimensions of the Psychosocial Impact of Assistive Device Scale (PIADS) and the Assistive Technology Device Predisposition Assessment (ATD PA) device form.

Independent Variable	General Results (*n* = 28)	The Survey was Answered by…			
		The Person without Help (*n* = 10)	The Person with Help of Caregiver (*n* = 3)	The Caregiver (*n* = 13)	The Professional (*n* = 2)
	Mean (ST)	Mean (ST)	Mean (ST)	Mean (ST)	Mean (ST)
Competence (PIADS)	0.60 (0.98)	0.75 (1.00)	0.86 (1.47)	0.42 (0.99)	0.58 (0.47)
Adaptability (PIADS)	0.56 (1.31)	0.98 (1.25)	1.06 (1.54)	0 (1.25)	1.33 (0.94)
Self-esteem (PIADS)	0.49 (0.91)	0.59 (0.43)	0.79 (1.71)	0.20 (0.92)	1.38 (1.24)
Score of matching (ATD PA)	3.94 (0.73)	3.96 (0.42)	3.87 (0.74)	3.82 (0.92)	4.67 (0.47)

**Table 3 ijerph-18-00763-t003:** Variables with significant influence on dimensions of the PIADS.

Categorical Variable ^1^		Competence			Adaptability			Self-Esteem		
		Mean (ST)	F	*p* Value ^2^	Mean (ST)	F	*p* Value	Mean (ST)	F	*p* Value
The device was requested by…	Affected person	0.91 (1.05)	0.75	0.132	1.13 (1.27)	**137.4**	**0.028**	0.88 (0.89)	**258.3**	**0.029**
	Caregiver	0.33 (0.89)			0.02 (1.20)			0.12 (0.83)		
	Professional	0.25			0.67			0.50		
Type of device	Rest	0.09 (0.96)	**365.2**	**0.011**	0.30 (1.12)	0.982	0.051	0.56 (0.78)	**284.3**	**0.022**
	Transfers	0.10 (0.72)			−0.50 (1.21)			−0.23 (0.99)		
	Mobility	0.83 (0.71)			0.79 (1.28)			0.32 (0.81)		
	Bathing	1.38 (0.65)			1.37 (1.00)			0.83 (0.33)		
	Communication	2.50			2.83		.	2.75		
			**t**			**t**			t	
Diagnosis	ALS	0.40 (0.92)	**−3.8**	**0.028**	0.28 (1.22)	**−4.1**	**0.018**	0.39 (0.97)	−1.7	0.24
	NMD	1.50 (0.77)			1.83 (0.97)			0.93 (0.38)		
**Quantitative Variables**		**Pearson Coefficient**		***p*** **Value**	**Pearson Coefficient**		***p*** **Value**	**Pearson Coefficient**		***p*** **Value**
Age		−0.450		**0.019**	−0.442		**0.021**	−0.159		0.427
Score of matching (ATD PA) ^3^		0.547		**0.003**	0.590		**0.001**	0.710		**0.000**

^1^*t*-test (for two groups comparison) and one way ANOVA (for three or more groups comparison) were employed; ^2^*p* value and F is marked with bold when the relationship is significant. ^3^ ATD PA: Assistive Technology Device Predisposition Assessment.

**Table 4 ijerph-18-00763-t004:** Models emerging from lineal regression for dimensions of the PIADS.

Variables	Model 1		Model 2		Model 3	
	Standardized β	*p* Value	Standardized β	*p* Value	Standardized β	*p* Value
**Lineal Regression for Competence ^1^**						
Type of AT	0.612	0.001	0.547	0	0.485	0
Score of matching (ATD PA)			0.486	0.001	0.436	0.001
Diagnosis					0.342	0.007
R^2^	0.374		0.606		0.716	
ΔR^2^	0.349	0.001	0.573	0	0.679	0
**Lineal regression for Adaptability ^2^**						
Score of matching (ATD PA)	0.59	0.001	0.52	0.001		
Diagnosis			0.41	0.009		
R^2^	0.35		0.52			
ΔR^2^	0.32	0.001	0.48	0		
**Lineal regression for Self-esteem ^3^**						
Score of matching (ATD PA)	0.71	0				
R^2^	0.57					
ΔR^2^	0.52	0				

^1^ The final model for competence was significant (F = 19.31, *p* < 0.01), and the maximal FIV was 1.07. ^2^ The final model for adaptability was significant (F = 12.75, *p* < 0.01), and the maximal FIV was 1.03. ^3^ The final model for self-esteem was significant (F = 26.37, *p* < 0.01), and the maximal FIV was 1.

## Data Availability

Data is contained within the article.

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
