# Peer review of "How Loan Bank of Assistive Technology Impacts on Life of Persons with Amyotrophic Lateral Sclerosis and Neuromuscular Diseases: A Collaborative Initiative"

_ijerph, 2021, doi:10.3390/ijerph18020763_

Round 1
Reviewer 1 Report
The sample was formed by people with rare neurodegenerative disorders, it is rare disease therefore recruited small sample size. The ethical considerations are well design and structured written. Only one comment, the abstract present sample is "people with rare neurodegenerative disorders", however, the paper title is "persons with Amyotrophic Lateral Sclerosis", any difference or should revised to be consistent.
Author Response
Thank you for your suggestion. We have changed the information in abstract to meet correctly to the participants in the study: people with Amyotrophic Lateral Sclerosis and Neuromuscular Disorders.
This change is reflected with tracked version in page 1, line 16.
Reviewer 2 Report
You did an excellent job on the revision.
Author Response
Thank you so much for your time to read and review the manuscritp and for the recommendations done that have contributed to enrich the final paper.
This manuscript is a resubmission of an earlier submission. The following is a list of the peer review reports and author responses from that submission.
Round 1
Reviewer 1 Report
This is a very interesting paper and provides a novel addition to our exisiting knowledge base.
This is a very interesting article that adds novel information to our knowledge base.
Is the ATD PA also available in Spanish. Has it been validated for use in Spain?
Line 185. The main type of AT requested was for rest (N=10), following by those for mobility (N= 7).
Please name the technologies in these two categories. Examples in the remaining categories should also be provided.
Line 202. In this case, the score obtained taking into account the individual that covered the survey is not remarkable, but it is a little higher when the professional answered it.
This is an interesting finding and worthy of more attention in the Discussion.
In Table 3, it is Pearson coefficient. Also, double check that ATD PA is abbreviated consistently.
Double-check the accuracy of the data in Table 3 with the narrative. The entire Results section needs to be reviewed and edited for clarity.
Lines 326-330. The two citations are dated and there is evidence in the literature that the PIADS and ATD PA have shown validity for all population groups and all types of products. Consider eliminating this paragraph, or re-writing it.
Line 341. Add personnel as it is also one of the 5 P’s.
Throughout the article, awkward English makes the reader have to read sentences and paragraphs multiple times to get to the meaning, and then it is not always a certain that occured. The article needs copyediting by a Native English speaker or a copyediting service. This includes the title of the article.
Author Response
The answers are written in green color

Reviewer 2 Report
This paper present a cross-sectional study to evaluate the impact of Assistive Technology (AT) devices on the life of people with rare diseases; the study involved 28 participants in the region of Galicia (Spain) with the collaboration of some Non-Profit Organisations and the "Loan Bank of Assistive Technology”. The main contribution of the paper is the implementation of outcome measurement tools to determine the impact of AT devices and to evaluate the matching between the person and technology.
The paper is well structured, the methodology adopted is clear and ample space has been devoted to discussing the results.
1) The title in not really clear; the title should outline the main contribution of the paper: the implementation of outcome measurement tools to access the impact of Assistive Technology, not to improve the Quality of Life.
2) "Quality of Life (QoL)" and "Well-being" are keywords used in the article that need to be clarified. Which measurement tools/
models exist in the literature to evaluate the QoL? Please, improve the introduction section with new references on the topic (Wellness assessment, Quality of Life model, QoL indexes, etc.).
2) Is there any relationship between the disease and the outcome measurement tools that have to be used? The relation between different tools and the main dimensions of the QoL model (e.g. Physical, Social, and Cognitive dimensions) should be investigated. Moreover, it would be interesting to evaluate the results of this study on other degenerative pathologies that affect the autonomy of the patient.
3) Although the study is limited to a very small population, the limitations of the study are clearly indicated in the paper.
Spell check required.
Author Response
The answers are written in green color.

Reviewer 3 Report
The objectives of this study was two-folded: 1) to determine the impact and benefits of the assistive devices (aka assistive technology; AT) provided by Galician Association of Amyotrophic Lateral Sclerosis (AGAELA), a non-profit organization (NGO) in Spain for their clients with Amyotrophic Lateral Sclerosis (ALS) and Neuromuscular Diseases (NMD), and 2) to evaluate the matching system between users and the device(s) offered by the loan bank of AGAELA. To achieve the first objective, the authors applied the self-report questionnaire, Psychosocial Impact of Assistive Technology Scale (PIADS), to measure the users’ 3 domains: competence, adaptability, and self-esteem. To achieve the second objective, the authors applied the battery of the Matching Person and Technology (MPT), to assess the predisposition of the users using the assistive device based on his/her goals, preferences, technological features, and environmental circumstance. This cross-sectional study showed that the impact and benefits of the device(s) loaned from AGAELA was moderately positive and identified that the correct match between users and the AT was the main factor while using the MPT-modeled matching system derived from the counseling of an Occupational Therapist professional, suggesting that the type of loaned device and the diagnosis serves a moderator for the positive impact of AT on the user.
The merit of developing an effective and efficient local AT loan-bank for the people with severe medical condition living in poverty and/or facing barriers that hinder their access to medical assistive devices should be advocated and promoted and should by no means be diminished. There is no doubt that this current study took efforts and initiative to follow this direction, this particular work, however, suffers from the lack of generalizability in narrative of the unclear statistical analyses used in the text and, perhaps, a potential conflict of interest between the author(s) of the manuscript and the AGAELA.
Lack of generalizability:
First, while the current title of this study offered a promising headline to readers regarding how to improve quality of life (QoL) through loan bank of AT, what was actually conducted in this study, however, was to obtain data from the 28 subjects who loaded AT through AGAELA using the PIADS, which was only one of measurements for assessing a person’s (QoL) score.
Second, the authors stated in Introduction that “The rare degenerative diseases, such as ALS and NMD, have an important impact on the independence of the affected person.” First of all, every degenerative diseases affect independence of the patient somewhat degree, regardless whether the condition is “rare”. According to Statista, in 2016 for example, the conditions by the number of affected individuals in Spain were ranked from top Toxic oil syndrome (14,859), Arnold-Chiari Syndrome (162), epidermolysis bulls dystrophic (129), to ALS (79). This study was concluded and discussed based on the collected and analyzed data from 28 subjects with only two rare degenerative diseases(23 ALSs; 5 NMDs) from AGAELA. One question would be raised, why only “rare degenerative disease, such as ALS and NMD”? The current text provided no explanation why these two dispositional conditions were the targets the study. Was the study designed based on a grounded rational, or because of the data availability at AGAELA? The authors should explicitly state in the Introduction why samples with “rare degenerative disorders”, in general, ALS and NMD, in particular, were important to be considered in this study and how it was coped with patient use of AT.
Third, this study targeted Spanish population, in general, AGAELA, in specific. The authors stated, in Line 83, that the application of these tools used in the current study was published previously in two research projects (see current citations #19 and #20). However, these publications were non-English publications. There was no neutral capacity for English-speaking individuals to provide a critical review and understanding from non-English publications where the application tools used in this study for an English-writing literature/journal. Also, the authors mentioned in Line 81: “In Spain, the research related to outcome measures in AT is deficient, because there are not many specific measurement instruments that are validated in the Spanish population. “What make the Spanish population unique? If this study is considered to publish on international and English-writing journal, this concern should be addressed in the very beginning.
Other questions that may be explored to improve generalizablity in the future may include but not not limit to:
- How this AGAELA data be implemented in a bigger scope, such as other conditions above-mentioned in Spain? Perhaps, Europe, and the rest of the world?
- We had already known, from many practices and literatures, with or without this work, that an Occupational Therapist professional plays an important role in counseling, and it is a straightforward logic that the type of loaned device and the diagnosis serves a moderator for the impact of AT on the user. The questions is how this matching system is effective and efficient when the sample of the study included 23 ALSs and 4 NMDs? Let alone these two conditions are easily classified.
Scientific Soundness based on statistics:
The authors stated, in Lines 169-275, that Pearson correlation, t- paired test and ANOVA were used to determine those possible relationships between variables, and that hierarchical linear regression analyses for the three dimensions of PIADS were performed after controlling for relevant demographic and clinical covariates. It is unclear when and what test was used for what purpose throughout the text. In Results, the authors only provided descriptive results and tables of means and p-values without explaining where the analyzed results came from what test for what purpose. For example, ANOVA mentioned in the text. What typic of ANOVA was used? The author claimed to use SPSS for data statistical analysis, which shall provide details of the test name with post-hoc - if there was any. I unfortunately have never encountered a scientific article conducting ANOVA tests without reporting significant F- and p- values for interaction and/or for main effect of a factor of interest. Interestingly, in this report, there were none reported. Furthermore, the authors used the same PIADS data set to perform the linear regression after controlling for relevant demographic and clinical covariates. Generally speaking, the linear regression is used for making prediction; whereas the Pearson correlation is used for positive/negative relationship between 2 variables. It was unclear why the authors used many tests: Pearson correlation, t- paired test, ANOVA and hierarchical linear regression analyses for the same set of data while the conclusions of the study (Abstract, Introduction, Lines from 388) lean toward Pearson correlation - as it seemed. It was also very confusing to read the results 
where the author claimed that “the dimension of PIADS best score was competence, and the variations to gender were not”, assuming that the authors had analyzed the effect of gender and only variations to gender was not significant, while the authors also conducted the linear regression analyses “after controlling for relevant demographic and clinical covariates”, assuming gender was not a main factor? Were there interactions between gender and the PIADS domains in the first place? Was there main effect on gender on all measurements? What test(s) to prove that? Please specify and explicitly explain in the text.
Also, it is noted that using the same set of dataset and performing multiple analyses decreases the power of significance in a given study. The authors should explain in the text how to overcome this concern.
Potential conflict of interest:
Generally speaking, A conflict of interest is “a situation in which a person or organization is involved in multiple interests, financial or otherwise, and serving one interest could involve working against another.“ It is clear that the second and the third authors were affiliated with AGAELA by the time the study was conducted. There was no mentioned to declare their roles and responsibilities at AGAELA, in the text, especially in Conflict of Interest section. Instead, in this section, the authors declared no conflict of interest and wrote, “The funders had no role in the design of the study; in the collection, analyses, or interpretation of data; in the writing of the manuscript, or in the decision to publish the results. All procedures performed in studies involving human participants were under the ethical standards of the Research Ethics Committee of Galicia and with the 1964 Declaration of Helsinki and its later amendments or comparable ethical standards. This article does not contain any studies with animals performed by any of the authors. Informed consent was obtained from all individual participants included in the study. “ This part should go to the Methods and Methods where ethical standards were mentioned, not the Conflict of Interest.
It is also interesting to observe, while there have been consistent, clear, and frequent mentions of AGAELA and a “local/regional NGO” referring “AGAELA” through the entire text, including the Acknowledgement, there was nowhere mentioned the name of another NGO in Spain where the included NMD subjects were associated in the text, not even credited in Acknowledgement.
Suggestions:
Other pursing publication options that are considered more appreciate may be the annual report of AGAELA where the users, care providers, and OT professional in the AGAELA community may benefit the most. Since the authors were making efforts in preparing the text in English for a boarder audience, another publication option that would benefit the public could be local publication or conference in Spain. It may open doors for people living in Spain who speak English.
Minor concerns include sentences that do not follow the rules of English grammar and errors are not limited to the following:
Lines 17: Spell out PIADs when abbreviation is initially introduced.
Lines 19: Name and define each dimension.
Lines 24: Use of abbreviations such as “AT”
Line 96: what month of 2014 was missing.
Line 97: “Before implementing the research, the boards of directors of both entities were consulted, and they fix the agreement, after getting approval by the Galician ethic committee, to performance the study.” What does “they (the boards of directors) “fix” the agreement” mean?
Line 112: “… where the two NGOs of people with rare diseases are working and offering the service of loan bank of assistive technology.”
Line 131: “Matching Person and Technology (MPT) – Questionnaire: Assistive Device Predisposition Assessment: The MPT is a model based on Person, Milieu, and Technology factors and it considers important aspects of these domains in the assessment through a process divided into several steps in a User-Focused AT Assessment. “
Lines 422: “Acknowledgments: The support received from the Research cCnter of ICT (www.citic-research.es), and from the Galician Association of Amyotrophic Lateral Sclerosis. ”
Author Response
The answers are written in green color.

Reviewer 4 Report
1.the research title can not reflect the study content which focus on the subject of people with rare diseases.
2.the study do not describe how to find the study subject instead just present the inclusion criteria. And, the representation of the study subject need to evaluate to generalize the quality of life in this study.
3.the study general validity and reliability need to describe in the content.
4.sample size is too small to generalize the quality of life to this group of people is the greatest limitation in this study.
5. PIAD is not equal to quality of life which is the vital outcome of this study, the author should discuss it in detail.
Author Response
The answers are written in green color.

Round 2
Reviewer 4 Report
Although the author have revised the manuscript thoroughly, there are still many limitations in this study, such as the limitation of very small population will difficult to generalize the results. The author still did not answer the quality of study validity and validity appropriately.